# The Impact of Element Ordering on LM Agent Performance

## Abstract

There has been a surge of interest in language model agents that can navigate virtual environments such as the web or desktop. To navigate such environments, agents benefit from information on the various elements (e.g., buttons, text, or images) present. However, it remains unclear which element attributes have the greatest impact on agent performance, especially in environments that only provide a graphical representation (i.e., pixels). Here we find that the *ordering* in which elements are presented to the language model is surprisingly impactful—randomizing element ordering in webpages compromises average agent performance to a degree comparable to removing all visible text from webpages. While web agents benefit from the semantic hierarchical ordering of elements available via the browser, agents that parse elements directly from pixels do not have access to any such ordering. Here we endeavor to derive effective orderings and investigate the impact of various element ordering methods in web and desktop environments. We find that dimensionality reduction provides a viable ordering for pixel-only environments. We train a UI element detection model to derive elements from pixels and apply our findings to an agent benchmark—OmniACT—where we only have access to pixels. Our method completes more than two times as many tasks on average relative to the previous state-of-the-art.

## 1 Introduction

There has been growing interest in using language model (LM) agents to autonomously navigate virtual environments. Autonomous web agents (Zhou et al., 2023; Kim et al., 2023; Zheng et al., 2024; Gur et al., 2024; He et al., 2024) have become a particularly popular area of research. Typically, a web agent takes as input a task prompt from a user, observes a text and visual representation of the environment, and then outputs one or more actions to execute the task in the environment. Recently, research interest has expanded to include agents that can navigate mobile (Rawles et al., 2023; Yan et al., 2023) and desktop (Xie et al., 2024; Kapoor et al., 2024; Bonatti et al., 2024) environments as well.

At a high level, a virtual environment consists of numerous elements—some are interactive (e.g. buttons or widgets), while others are not (e.g. plain text). To allow for human navigation, these elements are usually represented in the pixel space via a Graphical User Interface (GUI). In contrast, agents often rely on distinct state representations to navigate virtual environments. The exact format of a state representation varies between environments and approaches. In web environments, common text representations include the HTML or accessibility tree (Zhou et al., 2023; Koh et al., 2024). For visual representations, a popular approach is to label UI elements with bounding boxes and numeric identifiers (Koh et al., 2024; He et al., 2024), known as *Set-of-Mark* (Yang et al., 2023). In either case, the state representation is derived from the underlying Document Object Model (DOM) (Zhou et al., 2023; Koh et al., 2024; He et al., 2024). However, many environments lack a descriptive DOM and *only* provide pixel information, which we show is insufficient for existing agents (see Section 4.2). To construct an effective state representation from only pixels, it is important to answer the following basic questions about these representations: What aspects of a state representation are most important to an agent? How can we derive these important aspects with only pixels?

In almost all implementations of the agent's state representation, there exists a list of interactable or non-interactable elements which the agent uses to determine the next action (Koh et al., 2024; Zhou

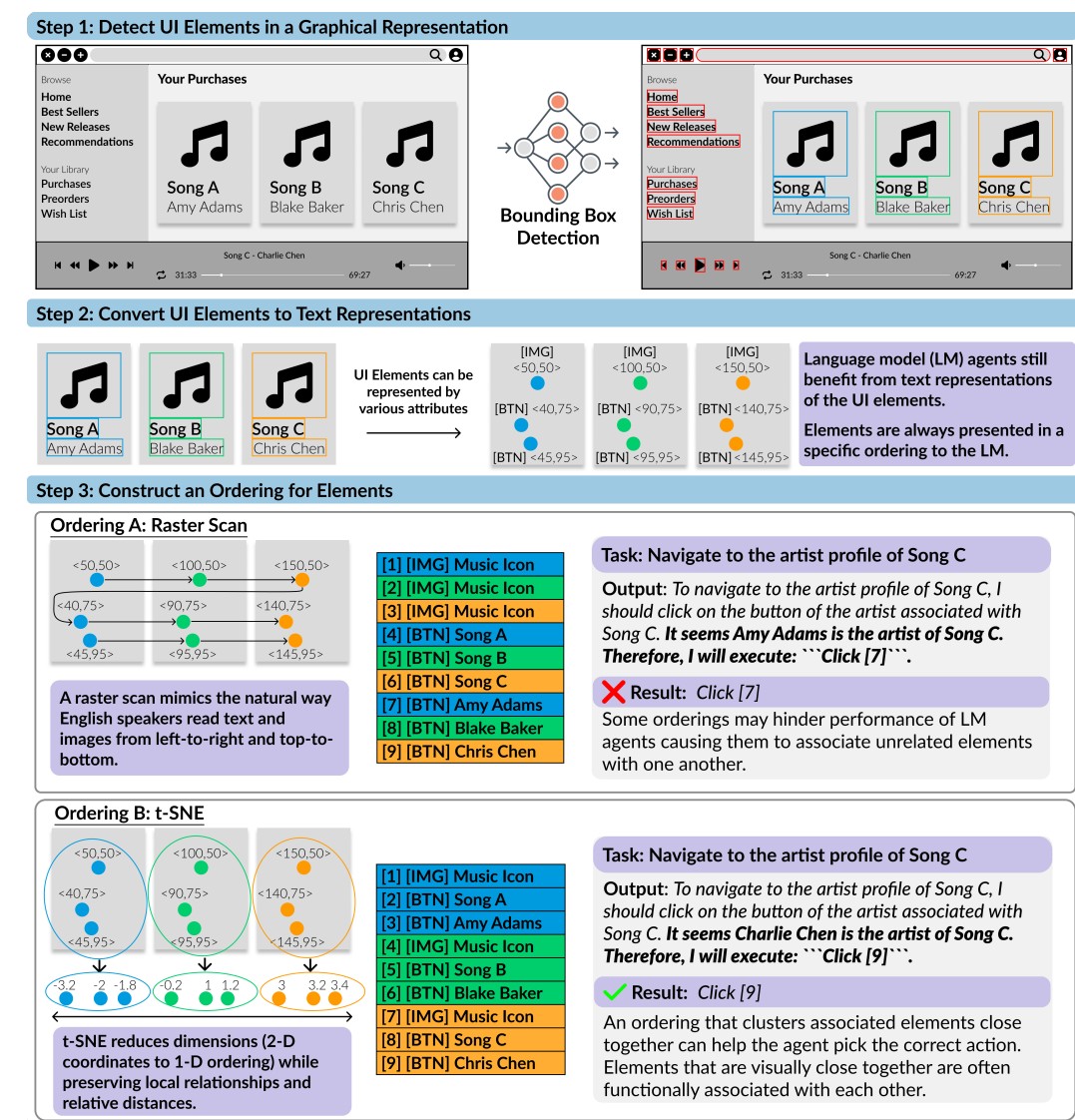

Figure 1: We are motivated by the goal of enabling agents to act on environments where an underlying DOM does not exist. Instead, the agent must determine its next action using only the environment's graphical representations. In Step 1, we first detect a list of unordered UI elements using an object detection model and identify them with bounding boxes. In Step 2, we convert these UI elements to their text representation. In Step 3, we order the elements via 2D-to-1D dimensionality reduction. Due to the sequential nature of a language model, elements are always presented in a specific order to the LM Agent. Finding an effective ordering is non-trivial, yet can significantly affect agent performance. Elements that are visually close together are often functionally associated with each other. t-SNE's ability to retain local structure allows it to generate an effective ordering.

et al., 2023; He et al., 2024; Dupont, 2024; Yan et al., 2023; Ishan0102, 2024; Kapoor et al., 2024; Xie et al., 2024). Elements are characterized by various attributes such as visual appearance, text descriptions, or type labels. Because the state representation is the input to an LM, this list of elements is always presented in a specific ordering. For example, the default method to derive elements from a webpage performs a pre-order traversal of the DOM tree (World Wide Web Consortium, 2013). We analyze various attributes of a popular state representation for agents and find element ordering to be the single most impactful attribute to agent performance. We find that the ordering of elements can dramatically affect the performance of an agent, resulting in differences of up to 49% relative performance.

Figure 2: Many software applications lack informative accessibility trees or DOMs. The accessibility tree for a popular game development engine (*Godot*, left) contains only the exit, minimize, and full screen buttons. For a presentation slide (*Google Slides*, right), no interactive elements (e.g. title and subtitle boxes) are present in the DOM. Language model agents rely on this information to navigate applications, and agent performance can accordingly be compromised in scenarios where it is incomplete.

This can prove problematic as many environments lack obvious methods to both derive and order elements. For example, many mobile and desktop applications (see Figure 2) do not properly expose interactable elements (Chen et al., 2020; Ross et al., 2020; Zhang et al., 2021). In such environments, pixels may be the only source of information available. Previous approaches to deriving interactable elements from pixels either leverage off-the-shelf segmentation models (Yan et al., 2023; Kapoor et al., 2024) or build custom models that target accessibility features (Wu et al., 2023). In our approach, we leverage common crawl (Common Crawl, 2023) to train an object detection model (Ren et al., 2016) that detects interactable UI elements specifically for agents. To the best of our knowledge, the elements detected through these approaches are ordered arbitrarily (e.g. based on confidence scores); visually, the ordering is effectively random. Our experiments indicate that a random ordering consistently results in the lowest performance across multiple scenarios.

Here we propose and evaluate strategies for deriving effective element orderings in scenarios where a hierarchical ordering based on the GUI design is not explicitly provided by the environment. Across multiple agent benchmarks, we find that ordering elements via a 2D-to-1D dimensionality reduction (Van der Maaten and Hinton, 2008) reliably yields improvements to agent performance relative to other baselines. We experiment on the VisualWebArena (Koh et al., 2024) and OmniACT (Kapoor et al., 2024) benchmarks and achieve new state-of-the-art performance on OmniACT.

Out contributions are as follows.

- We conduct a thorough ablation of VisualWebArena's state representation for agents by including or removing each element attribute individually. Despite advancements in vision language models, we find that a text representation is still necessary for web and desktop agents. We find that element ordering is, perhaps surprisingly, more impactful than any other attribute in the text representation.

- We demonstrate that ordering via dimensionality reduction consistently provides performance improvements over random ordering. Additionally, we find that ordering via dimensionality reduction performs better than a simple position-based ordering in most scenarios.

- We achieve a new state-of-the-art result on OmniACT, an agent benchmark that considers the scenario of operating on pixels. Our approach more than doubles the expected average task success rate compared to the previous state-of-the-art.

## 2 RELATED WORK

**Agent Benchmarks.** World-of-bits provided the first environment for evaluating web GUI navigation using an agent (Shi et al., 2017). Over time, more realistic web (Zhou et al., 2023; Koh et al., 2024; Yao et al., 2023), desktop (Kapoor et al., 2024; Xie et al., 2024), and mobile (Rawles

et al., 2023) agent benchmarks have been created. Kim et al. (2023) provided one of the first LM agent approaches, successfully navigating World-of-bits. However, existing agents are still unable to properly navigate more realistic benchmarks, completing only 15% of web tasks (Zhou et al., 2023) and 12% of desktop tasks.

**Agents With Direct Access to Elements.** Despite work on multimodal agents (Koh et al., 2024), existing techniques in navigating web and desktop environments still rely heavily on ground-truth text representations. Zhou et al. (2023); Koh et al. (2024) both utilize the accessibility tree and its elements as their state representation. Koh et al. (2024); He et al. (2024) consider approaches where interactable elements are also represented in an image via Set-of-Mark (Yang et al., 2023) bounding boxes and labels. Xie et al. (2024) provides an agent that navigates desktop applications by observing a filtered down version of the accessibility tree. All of these approaches require access to either a webpage's underlying DOM or an accessibility tree to derive elements; however, our focus is on environments that only give access to their graphical representations which is significantly more challenging.

**Agents With Access to Only a Graphical Representation.** There have been several approaches— primarily focused on desktop and mobile environments—to directly navigating a GUI via its pixels. Kapoor et al. (2024) and Yan et al. (2023) focus on navigating desktop and mobile applications respectively. Both leverage an off-the-shelf-segmentation model—Segment Anything (Kirillov et al., 2023)—to find icons in the image. These icons are then either represented in text (Kapoor et al., 2024) or labeled with Set-of-Mark (Yang et al., 2023) bounding boxes and labels in the image (Yan et al., 2023). We instead train an object detection model that detects interactable UI elements directly. While previous UI element detection models are trained to detect accessibility features (Wu et al., 2023), our model is trained specifically to detect interactable elements that would be useful to an agent. All three of our approaches use Optical Character Recognition (OCR) modules such as EasyOCR (AI, 2020) to extract text from pixel information.

**Agent Input Ablations.** While most agent studies include some ablations, few focus on detailed analysis of an agent's input. To our knowledge, Huq et al. (2023) is the only other study that directly studies this. Their study focuses on broader components to an input prompt such as the selection of few-shot examples used, while we focus on specific element attributes such as element ordering.

## 3 PROBLEM DEFINITION

We define the environment state $\mathcal{E}$ as a set of elements $\mathcal{E} = \{e_1, e_2, \ldots, e_n\}$, where each element $e_j$ is a tuple $\langle i_j, C_j, A_j, S_j \rangle$ defined by the following parameters:

- $i_j \in \{0, 1\}$ denotes the interactability of element $e_j$. An element with $i_j = 1$ is interactable, while an element with $i_j = 0$ is not.
- $C_j = \{\langle x_1, y_1 \rangle, \langle x_2, y_2 \rangle\}$ is the set of pixel coordinates that form the bounding box around $e_j$. $\langle x_1, y_1 \rangle$ is the top left coordinate and $\langle x_2, y_2 \rangle$ is the bottom right coordinate.
- $A_j = \{a_1, a_2, \ldots, a_m\}$ is the set of potential actions that can be taken on that element. For example, the potential actions for a search bar might be $\{\texttt{click}, \texttt{type}\}$; a non-interactive text element has the action set $\varnothing$.
- $S_j$ represents the set of other environment-specific attributes for element $e_j$. These attributes can include image captions, type labels (e.g., Button, Text Field), or accessibility information.

While certain environments may provide full access to this environment state, here we are focus on environments where only the pixel information, $\mathcal{P}$, is available. We then must predict elements from the pixel information to construct a state representation for the agent. In other words, we must find a function $g \colon \mathcal{P} \to \mathcal{E}$

Elements can be represented in both visual and text modalities. For images, the most common approach is to overlay bounding boxes with numeric identifiers around each interactable element (Koh et al., 2024; He et al., 2024; Yan et al., 2023) in a manner inspired by Set-of-Mark Prompting

(Yang et al., 2023). In text, a common approach is to represent each element as a string containing its index, coordinates, and other attributes, such as "[1] [x,y] [Description]". Because LMs operate on sequential data, elements must be given an ordering; in most approaches, this is implicitly defined by the method used to identify the elements.

**Ordering.** The ordering is defined as a permutation $\sigma$ of the indices $\{1, 2, \ldots, n\}$ where $\{e_{\sigma(1)}, e_{\sigma(2)}, \ldots, e_{\sigma(n)}\}$ represents a specific sequence of elements. An ordering function is a function $f \colon \mathcal{E} \to \sigma$ that takes the environment as an input and yields a specific ordering $\sigma$.

# 4 WHICH ASPECTS OF AGENT STATE REPRESENTATIONS ARE MOST IMPACTFUL?

Here we describe a series of ablation experiments designed to examine which aspects of an LM state representation are most impactful to the performance of LM agents. In particular, we experiment on the VisualWebArena (VWA) (Koh et al., 2024) benchmark and ablate attributes of the state representation of the state-of-the-art agent (proposed in the same paper). We pick this representation in particular due to both its strong performance on VWA, as well as its similarity to common practices seen in other agent research (He et al., 2024) and open source projects (Ishan0102, 2024; Dupont, 2024). Our experiments in turn ablate the impact of (1) multimodal (image and text) aspects of the state representation, and (2) individual element attributes within the text component alone.

## 4.1 VISUALWEBARENA

VisualWebArena focuses on multimodal tasks in the web and provides a self-hostable environment for language agents to navigate (Koh et al., 2024). Agents operating on VisualWebArena have full access to the DOM. The current best approach (Koh et al., 2024) utilizes a multimodal representation where elements are parsed in a pre-order traversal of the DOM tree (World Wide Web Consortium, 2013). Each element $j$ has attributes $S_j = \{id, tag, text\}$ where $id$ is a unique numeric identifier for interactable elements and $\varnothing$ otherwise, $tag$ is the HTML tag (e.g. BTN or IMG), and $text$ is the alt text and captions for images and the HTML text otherwise. In the text representation, an example of an element would be "[1] [IMG] [alt text, caption]". In the image representation, each element is labeled with bounding boxes and numeric labels.

The original agent in (Koh et al., 2024) only achieves 15% success rate across all tasks. Since our goal is not to improve agent performance on VisualWebArena but rather to understand the importance of attributes in the state representation, we examine a subset of tasks to reduce costs.[0] Specifically, we explore tasks marked as "easy" within tasks that the original agent completed successfully. Due to variance associated with stochastic LM outputs, our reproduction of these originally-successful tasks yields a success rate around $74.07\% \pm 5.56\%$. The exact list of tasks can be found in Appendix A.8. We reuse the action space from the original agent which consists of executing high-level actions (e.g., click, hover) on individual elements—see Appendix A.6 for more details.

## 4.2 ABLATION SETUP AND FINDINGS

The agent state representation we explore is multimodal and consists of image and text information. The image consists of a screenshot of the webpage along with Set-of-Mark annotations, while the text consists of a DOM-ordered list of elements with the attributes outlined above. Our ablation protocol consists of removing individual attributes from the image or text representation and measuring task success rate—we say an attribute has high "impact" if its removal leads to substantial reduction in task success rate. To provide evidence that these findings may be robust across different LM backbones, we explore both GPT-4V as used in the original agent, and Gemini 1.5 Pro. Some experiments were not run on GPT-4V due to high associated costs, though we found ordering to be consistent between these two LM backbones on all ablations where we ran both.

In Table 1, we report results ablating aspects of the multimodal representation. In Table 2, we report the impact of ablating various attributes in the text representation specifically. Across all experiments, we consider the pre-order traversal of the DOM tree as the ground truth element ordering, and define

---

[0]A full run of the state-of-the-art agent on VisualWebArena can cost up to $800 with GPT-4V.

Table 1: Ablating the multimodal aspects of state representation in VisualWebArena. ✓ indicates ground truth obtained from the HTML. ✗ indicates removal of the attribute. We find that removing the text representation can dramatically harm agent performance.

| | Observations | | Success Rate (↑) | |
|---|---|---|---|---|
| Screenshot | Set-of-Mark | Text Representation | Gemini 1.5 | GPT-4V |
| ✓ | ✓ | ✓ | 64.20% | 74.07% |
| ✗ | ✗ | ✓ | 46.30% | 38.89% |
| ✓ | ✗ | ✓ | 50.00% | - |
| ✓ | ✓ | ✗ | 3.70% | 38.89% |

the "removal" of ordering information as substituting an ordering $\sigma_{rand}$ picked uniformly at random from all possible permutations. We summarize a few key findings from both sets of ablations below.

**Text Representation is Still Necessary.**    While adding a visual representation clearly improves performance, we find that it alone is insufficient even with Set-of-Mark labels. This contradicts previous findings on agents for mobile applications which found that a screenshot with Set-of-Mark labels achieves similar performance with or without text (Yan et al., 2023). We speculate that this is due to the substantial difference in viewport sizes between mobile and desktop environments. Specifically, the average mobile device has a viewport size of 360x800 while the average desktop has a viewport size of 1920x1080 (Statcounter Global Stats, 2024). Additionally, larger viewport sizes have been shown to improve agent performance in desktop environments (Xie et al., 2024). We speculate that this may be because current agents almost never understand when to change the screen view (e.g. by scrolling).

**Removing Ordering Information Harms Performance More Than Removing Any Other Attribute.**    Although most element attributes are important, we find that ordering is the single most important attribute for agent performance. Random ordering results in a similar performance drop to removing all HTML text descriptions.

**Captions Impact Performance More Than Alt Text.**    Removing captions causes a greater decrease to performance than removing alt text. From our experience, captions almost always provide more information than alt text. In fact, captions frequently include the alt text directly in its description.

Table 2: Ablating attributes of the text component of the VisualWebArena state representation. All results include the screenshot with Set-of-Mark bounding boxes and labels. TAG is the HTML tag. CAPTIONS are image captions generated using BLIP-2-T5XL(Li et al., 2023). $\text{TEXT}_{Alt}$, $\text{TEXT}_{Interact}$, and $\text{TEXT}_{Static}$ are the alt text, text for interactable elements, and text for non-interactable elements respectively. ORDER is element ordering. ✓ indicates ground truth obtained from the HTML. ✗ indicates removal of the attribute. ✗ Element Ordering indicates a random shuffling of the elements. - denotes experiments that were not run due to cost.

| Element Attributes | | | | | | Success Rate (↑) | |
|---|---|---|---|---|---|---|---|
| TAG | CAPTIONS | $\text{TEXT}_{Alt}$ | $\text{TEXT}_{Interact}$ | $\text{TEXT}_{Static}$ | ORDER | Gemini 1.5 | GPT4-V |
| ✓ | ✓ | ✓ | ✓ | ✓ | ✓ | 64.03% | 74.07% |
| ✗ | ✓ | ✓ | ✓ | ✓ | ✓ | 61.11% | 61.11% |
| ✓ | ✗ | ✓ | ✓ | ✓ | ✓ | 46.30% | - |
| ✓ | ✓ | ✗ | ✓ | ✓ | ✓ | 68.15% | 66.67% |
| ✓ | ✓ | ✓ | ✗ | ✓ | ✓ | 53.70% | - |
| ✓ | ✓ | ✓ | ✓ | ✗ | ✓ | 57.40% | - |
| ✓ | ✓ | ✓ | ✗ | ✗ | ✓ | 35.18% | - |
| ✓ | ✓ | ✓ | ✓ | ✓ | ✗ | 37.04% | 44.44% |

## 5 EXPERIMENTAL SETUP

For the remainder of this paper, we leverage the insights gained from our state representation ablations on VisualWebArena to tackle a more challenging task: enabling LM agents to act in environments that only expose pixel information.

Most applications are built on top of an underlying hierarchical representation. For example, a webpage is modeled by the DOM which is hierarchical. When exposed, this hierarchy can be used to determine a strong element ordering. However, the availability and quality of an underlying hierarchical representation can vary greatly between environments. For example, Chen et al. (2020) found that 77% of mobile applications had missing labels and Ross et al. (2020) found that 53% of image buttons had missing labels and were incorrectly sized. Additionally, Ross et al. (2020) found that 8% of applications lacked interactable element information altogether. In such scenarios, we may only have access to the application's pixel information. We continue to experiment with VisualWebArena and also experiment with the OmniACT benchmark as a scenario where we only have access to pixel information. Details on the LM agent backbones used in our experiments can be found in Appendix A.7.

### 5.1 OMNIACT

OmniACT provides both web and desktop environments for agents to benchmark on. OmniACT contains 177 application screenshots overall and 2021 tasks in the test set. Agents are tasked with generating pyautogui code that can navigate the application screenshot. We consider OmniACT as a setting where only a pixel information is available.

To detect UI elements $\{e_1, e_2, \ldots, e_n\}$ when given only pixel information $\mathcal{P}$ we train an object detection model (Ren et al., 2016) to detect interactable UI elements in the screenshot and use EasyOCR (AI, 2020) to extract visible text. In other words, the function $g \colon \mathcal{P} \to \mathcal{E}$ is defined by the trained object detection model. We add visible text and captions to each interactable UI element. We gather a dataset by finding 674,416 interactable elements over 1468 Common Crawl webpages. We selected our webpages based on top websites from Similarweb (2024). Despite the domain shift from webpages to desktop applications, we found that our object detection model worked reasonably well on the OmniACT benchmark in the end-to-end agent setting. We publicly release this model along with our paper. Training details can be found in the Appendix A.1.

OmniACT provides partial human annotations for each screenshot; multiple, but not all, interactable UI elements are annotated with bounding boxes. The original intent of these bounding boxes is for evaluation only. As a result, there are significantly less UI elements annotated compared to possible UI elements in the application. We experiment with elements derived from these human annotated bounding boxes to understand a) impacts to ordering performance in an easier setting and b) the potential performance that can be gained by improving UI element detection.

### 5.2 METRICS

Our primary metric is task success rate which is the standard for agent evaluations (Koh et al., 2024; Zhou et al., 2023; He et al., 2024). VisualWebArena provides an evaluation framework for task success. Task success criteria include achieving an expected final webpage state or receiving a desired response from the language agent. OmniACT does not provide task success rate directly, instead introducing the sequence score and action score metrics. Sequence score evaluates if the output contains the correct high level action (e.g. `click` or `type`), but does not check if the action element or parameter (e.g. `click [1]` or `type [parameter]`) are correct. Action score evaluates if the output contains both the correct high level action and the correct element or parameter. Thus, for OmniACT we focus our evaluations on the action score as it is more similar to task success rate.

### 5.3 ORDERING METHODS

In addition to random ordering, we experiment with two different ordering methods.

**Random.** We pick the ordering $\sigma_{rand}$ uniformly at random from all possible permutations. This provides a baseline performance.

**Raster.**   Elements $\{e_1, e_2, \ldots, e_n\}$ are ordered in a left-to-right raster scan. We define a raster scan as an ordering $\sigma_{raster}$ where $i < j$ iff $\lfloor \frac{y_i}{8} \rfloor < \lfloor \frac{y_j}{8} \rfloor$ and $x_i < x_j$. We chose to discretize the scan to prevent jumps in ordering from minor pixel variations. This method mimics the natural way English speakers read text and images from left-to-right and top-to-bottom.

**t-SNE.**   We apply dimensionality reduction techniques to better capture the spatial relationships. Using t-SNE (Van der Maaten and Hinton, 2008), we reduce the dimensionality with the function $g\colon \langle x, y \rangle \to z$. The set of values $Z = \{z_1, z_2, \ldots, z_m\}$ is used to determine the ordering $\sigma_{tsne}$. We use the scikit-learn (Pedregosa et al., 2011) implementation of t-SNE and keep the default parameters.

Our intuition for choosing t-SNE stems from visually and qualitatively inspecting our agent trajectories. We observed that agents often use adjacently ordered elements as context clues to determine the correct action. Furthermore, agents have difficulty reasoning about functionally associated elements that are separated from each other in the ordering. t-SNE generates an effective ordering as elements that are visually close together (i.e. nearby in 2D pixel space) are often functionally associated with each other as well. When reducing dimensions, t-SNE retains local structure which increases the odds that functionally associated elements are adjacent in the induced 1D ordering.

## 5.4   ACTION SPACE

We use the same high-level action space as described in OmniACT. Unlike OmniACT, we do not have the model directly output pyautogui code. Instead, we use high level actions that map to pyautogui code. For example, for element $e_1$ defined by $\langle 1, \{\langle 50, 50 \rangle, \langle 100, 100 \rangle\}, \{\texttt{click}\}, \varnothing \rangle$ the model would output `click [1]` which would be converted to `pyautogui.click(75, 75)`. This prevents the model from having to reason about pixel coordinates directly. Each element's unique identifier reflects the position of the element in the ordering. The full action space is in Table 3.

Table 3: The set of possible actions in OmniACT.

| Action | Description |
| --- | --- |
| Click | Perform a single click on an element. |
| Double Click | Perform a double click on an element. |
| Right Click | Perform a right click on an element. |
| Move/Hover | Move the cursor over an element. |
| Drag | Click and drag an element to a new position. |
| Scroll | Scroll up or down the page. |
| Horizontal Scroll | Scroll left or right on the page. |
| Press | Press a key on the keyboard. |
| Keyboard Hotkey | Use a keyboard shortcut or hotkey. |
| Write | Type text using the keyboard. |

## 6   RESULTS

Our main findings on the impact of ordering are in Table 4. We utilize our various findings to improve upon OmniACT; our experiments against their baseline are in Table 5.

## 6.1   IMPACT OF ORDERING

**Ordering Consistently Impacts Performance.**   Ordering has a significant impact to performance across all of our experiments. Random ordering decreases performance in VisualWebArena by 50% and 42% relative performance for GPT-4v and Gemini-1.5 respectively. In all experiments, random ordering decreases performance over a proper ordering method.

**t-SNE Best For Larger Models And More Challenging Tasks.**   Navigating by using detected elements is a harder task than navigating by using human annotated bounding boxes; not only are there more elements—on average double the amount—there is the possibility that the correct element is missing from the detected elements. We see that when elements are derived from the DOM and our UI detection model, t-SNE ordering generally outperforms raster ordering. Additionally, more powerful models see an increased benefit from t-SNE ordering with Gemini-1.5 and GPT-4v seeing larger improvements than LLama3.

Table 4: Performance of different ordering methods across various models and information scenarios. The baseline approach for VisualWebArena is the same as their paper. Human annotations are from OmniACT's annotation files. The Faster-RCNN model is trained to detect interactable UI elements from CommonCrawl webpages. VisualWebArena is evaluated on success rate. OmniACT is evaluated on unweighted action score (i.e. each task is weighted equally). We use Llama3-70B for VisualWebArena and Llama3-8B for OmniACT due to its large size[2]. GPT-4v is evaluated on a 100 random tasks for OmniACT; the exact list is in Appendix A.8. Gemini-1.5 and GPT-4v are multimodal while Llama3 is text only.

| Experimental Settings | | | Success Rate | | |
|---|---|---|---|---|---|
| Element Source | Benchmark | Ordering Method | Gemini-1.5 ($\uparrow$) | GPT-4v ($\uparrow$) | Llama3 ($\uparrow$) |
| Ground Truth (DOM) | VWA | Pre-order | 64.03% | 74.07% | 27.79% |
| | | Random | 37.04% | 37.04% | 20.37% |
| | | Raster | 38.88% | 53.70% | 29.63% |
| | | t-SNE | 44.44% | 61.11% | 24.07% |
| Human Annotations | OmniACT | Random | 57.29% | 61.52%[*] | 28.67% |
| | | Raster | 61.04% | 65.88%[*] | 33.65% |
| | | t-SNE | 59.17% | 62.11%[*] | 31.99% |
| Detected (Faster-RCNN) | OmniACT | Random | 39.59% | 44.63%[*] | 18.88% |
| | | Raster | 45.21% | 47.38%[*] | 21.58% |
| | | t-SNE | 47.16% | 49.18%[*] | 24.61% |

**Raster Ordering Performs Best With Human Annotations.** Raster ordering performs the best with human annotated elements. Unfortunately, these annotations are fewer and partially guaranteed to contain important information. Additionally, high quality human annotations are difficult to scale across applications.

## 6.2 STATE-OF-THE-ART PERFORMANCE ON OMNIACT

We achieve a new state-of-the-art performance on OmniACT. Due to cost, we look to our previous experiments to pick the best combination of features for our approach. We observe that multimodal representations are still helpful. We find that t-SNE ordering improves performance the best in OmniACT when elements are detected by our model. Koh et al. (2024) states that high level actions are easier for an LM to reason with.

We analyze the differences between our best approach and OmniACT's baseline. These are as follows.

- **Element Source:** OmniACT obtains elements by searching for icons with Segment Anything (Kirillov et al., 2023) and text with EasyOCR. Unfortunately, there are no shared artifacts for their icon detection system. We obtain UI elements through an object detection model and text with EasyOCR (AI, 2020).

- **Ordering:** It is unclear how elements are ordered in OmniACT. Considering how most approaches don't pay specific attention to ordering, we assume the ordering in OmniACT is effectively random. We order our elements using our t-SNE ordering.

- **Action Space:** OmniACT directly outputs pyautogui code as their actions. We consider a higher level action space that maps to pyautogui code.

- **Intractability** OmniACT lists out each element, but does not indicate which element is interactable. We specify whether elements are interactable or not.

- **Multimodal Representation:** OmniACT evaluates their full test set using a text only representation[3]. We experiment with a multimodal representation.

We apply our findings and achieve more than one-fold increase over the existing best action score. Our results can be found in Table 5.

---

[2]We use the Groq API for our Llama3 models.

[3]OmniACT evaluates the impact of adding a visual representation on smaller subset; however, this subset is not shared and varies empirically from the full test set.

Table 5: ✓ and ✗ indicates whether the feature is available when building a representation for the model. Ours indicates a high level actions such as `click [1]`. Code indicates that pyautogui code is directly generated. **bold** is best and *italics* are second best. While sequence score only checks for the correct high level action (e.g. `click`), action score checks for both the correct action and the correct element or parameter (e.g. `click [1]`). Thus, action score is the most equivalent to task success rate. * is as reported in Kapoor et al. (2024)

| Model | State | Actions | Screenshot | Action Score (↑) | Sequence Score (↑) |
|---|---|---|---|---|---|
| GPT-4 | OmniACT | Code | ✗ | 11.60* | **32.75*** |
| Gemini 1.5 | Ours | Code | ✗ | 16.53 | 21.67 |
| Gemini 1.5 | Ours | High Level | ✗ | 22.29 | 29.42 |
| Gemini 1.5 | Ours | High Level | ✓ | *22.86* | 28.91 |
| Llama8b | Ours | High Level | ✗ | 18.64 | 26.22 |
| GPT-4v | Ours | High Level | ✓ | **23.34** | *30.47* |

## 7 FUTURE WORK

**Further Improvements to Ordering** We provided two simple methods to apply ordering when a default ordering is not given. However, both approaches still fall short when compared to the hierarchical ordering derived from the DOM. We hope that future research can introduce more sophisticated methods to find ordering with only pixel information.

**Image Only Ordering** We focused on the impact of element ordering in a text representation (although element labels re-ordered accordingly in the visual representation as well). The impact of element ordering may or may not generalize to an image only representation. Unfortunately, our results indicated that a visual representation alone was insufficient for web and desktop environments which prevents us from conducting this experiment. However, Yan et al. (2023) found that a visual representation in mobile environments was able to achieve comparable performance with and without a text representation. In the future, we hope to experiment with various ordering methods on an image only representation.

**Expanded Scenarios and Benchmarks** In this paper, we explored two benchmarks— VisualWebArena and OmniACT—as web and desktop scenarios. In the future, we hope to explore other benchmarks and settings with our approach. For example, Xie et al. (2024) uses the OS level accessibility tree for desktop agent navigation. We hope to compare our approach against theirs and believe that combining both approaches may lead to further improvements. Additionally, mobile environments often have only pixel-level information (Chen et al., 2020; Ross et al., 2020); we hope to apply our approach to a mobile benchmark such as Rawles et al. (2023).

## 8 CONCLUSION

We conducted thorough ablations to show that element ordering has a significant impact on the performance of agents. We provided a method of ordering elements through dimensionality reduction and showed that it performed best in realistic environments. We trained a UI element detection model on Common Crawl data and publicly share the model. We demonstrated an end-to-end method which allows a LM agent to act on environments that provide only pixel information. Using this method, we were able to achieve a new state-of-the-art performance on OmniACT.

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

# A APPENDIX

## A.1 FASTER-RCNN TRAINING DETAILS

We trained our model using the detectron2 Wu et al. (2019) implementation of faster-rcnn. We did not change much from the default implementation and recognize that there are significant improvements that could be made to the model.

Table 6: Key hyperparameters for the Faster-RCNN model.

| Hyperparameter | Value |
|---|---|
| Base Learning Rate | 0.00025 |
| Number of Classes | 1 |
| Iterations | 200000 |
| Optimizer | SGD |
| Backbone | Resnet-50 (ImageNet Pretrained) |
| ResNet Depth | 50 |
| Images per Batch | 16 |
| Objects per Image | 128 |
| Devices | 8 |

We share the remaining hyperparmeters in a config file. We also share the model artifacts and dataset in our github.

## A.2 FASTER-RCNN EVALUATION

We evaluated our detection results using the standard COCO evaluation metrics Lin et al. (2014), which include mean Average Precision (mAP) across IoU thresholds from 0.50 to 0.95, as well as performance breakdowns across different object scales.

Table 7: Object detection performance metrics across different dataset splits using out Faster R-CNN Ren et al. (2016) model. AP represents Average Precision, with subscripts denoting IoU thresholds. $AP_S$, $AP_M$, and $AP_L$ represent performance on small ($< 32^2$ pixels), medium ($32^2$ to $96^2$ pixels), and large ($> 96^2$ pixels) objects respectively. All values are percentages (%).

| Split | $AP_{50:95}$ | $AP_{50}$ | $AP_{75}$ | $AP_S$ | $AP_M$ | $AP_L$ | $AR_{100}$ |
|---|---|---|---|---|---|---|---|
| Train | 68.94 | 77.53 | 73.08 | 63.61 | 73.95 | 62.55 | 73.40 |
| Validation | 69.40 | 77.51 | 73.18 | 62.94 | 75.04 | 62.16 | 73.80 |
| Test | 66.87 | 75.47 | 70.96 | 64.77 | 70.16 | 62.10 | 71.30 |

## A.3 DATASET DETAILS

We built our dataset by automatically annotating Common Crawl webpages. We split the dataset into a randomized 70/20/10 (train/dev/test) split.

Table 8: Dataset statistics for our object detection task. Areas are measured in pixels squared.

| Metric | Value |
| --- | --- |
| Number of Images | 12,965 |
| Total Annotations | 674,416 |
| Average Annotations (i.e. Boxes) per Image | 52.02 |
| Max Annotations (i.e. Boxes) in One Image | 288 |
| Small Objects ($< 32^2$ px) | 18.4% |
| Medium Objects ($32^2$-$96^2$ px) | 62.2% |
| Large Objects ($> 96^2$ px) | 19.4% |

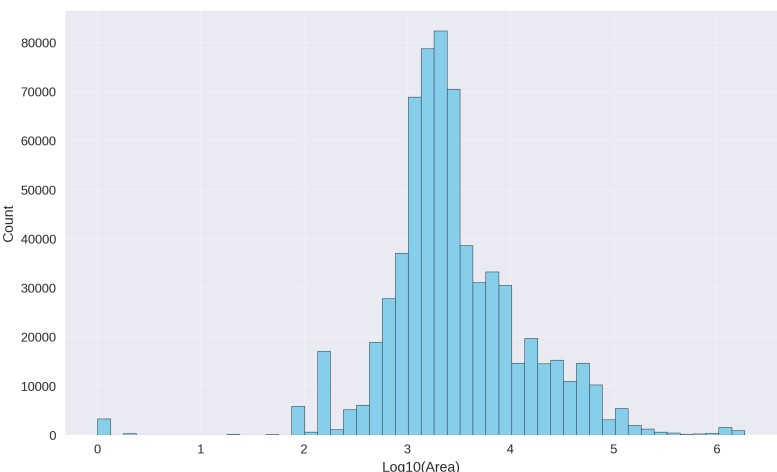

Figure 3: Distribution of bounding box areas in our dataset on a logarithmic scale. The x-axis shows the $\log_{10}$ of the area in pixels squared, while the y-axis shows the frequency.

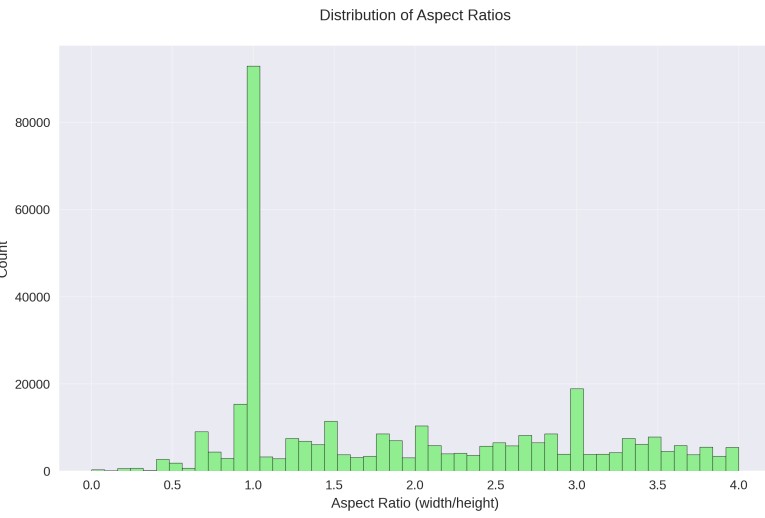

Figure 4: Distribution of bounding box aspect ratios (width/height) in our dataset. The x-axis is limited to ratios between 0 and 4 to focus on the most common aspect ratios.

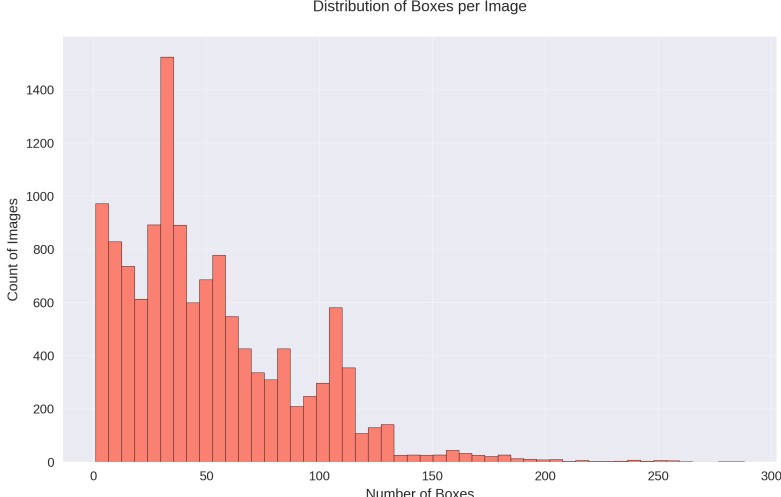

Figure 5: Distribution of the number of bounding boxes per image. This histogram shows the density of annotations across our dataset.

### A.4   EFFECTS OF ELEMENT COUNT ON ORDERING

We notice that the relative impact of element ordering becomes more significant as interfaces grow in complexity (Table 9). This affects both raster and t-SNE orderings similarly.

Table 9: Difference in task success rate compared to random baseline for Raster and t-SNE across different ranges of elements in the interface. The split is set to the median number of elements observed across VisualWebArena and OmniACT datasets.

| Element Count | Raster | t-SNE |
|---|---|---|
| 0-25 | +2.8% | +3.2% |
| 25+ | +6.3% | +6.5% |

### A.5   T-SNE HYPERPARAMETERS

We evaluated t-SNE across various perplexity values. In Table 10 we can see that varying perplexity values has a neglible effect to overall performance.

Table 10: Action scores achieved by the model across different t-SNE perplexity values.

| Perplexity | 10 | 20 | 30 | 40 | 50 |
|---|---|---|---|---|---|
| Action Score | 18.25 | 17.67 | 17.95 | 18.37 | 18.47 |

## A.6 VISUALWEBARENA AGENT ACTION SPACE

We use the same action space as VisualWebArena for all VisualWebArena experiments.

We use the same action space as described in VisualWebArena. VisualWebArena uses high level actions that act directly on elements rather than pixel coordinates. Interactable elements possess a unique `id` identifier while non-interactable elements do not. The `id` identifier reflects the position of the element in the ordering. The full action space is in Table 11.

| Action | Description |
|---|---|
| `click [id]` | Click on element `id`. |
| `hover [id]` | Hover on element `id`. |
| `type [id] [text]` | Type text on element `id`. |
| `press [key_comb]` | Press a key combination. |
| `new_tab` | Open a new tab. |
| `tab_focus [index]` | Focus on the i-th tab. |
| `tab_close` | Close current tab. |
| `goto [url]` | Open `url`. |
| `go_back` | Click the back button. |
| `go_forward` | Click the forward button. |
| `scroll [up\|down]` | Scroll up or down the page. |
| `stop [answer]` | End the task with an optional output. |

Table 11: The set of possible actions in VisualWebArena.

## A.7 LM AGENT HYPERPARAMETERS AND SETTINGS

We set our temperature, top-p, and input token limits based on existing works Zhou et al. (2023); Koh et al. (2024); Kapoor et al. (2024). Prompts for all three backbones contain few-shot examples and use chain-of-thought prompting Wei et al. (2023) as described in Koh et al. (2024). In GPT-4v and Llama3 each example is a different message; Gemini-1.5's context length allowed us to input all examples as a single prompt. We detail our LM Agent hyperparameters in Table 12

| Setting | Language Model Backbone | | |
|---|---|---|---|
| | GPT-4v | Gemini-1.5 | Llama3 |
| Input Token Limit | 3840 | 900000 | 3840 |
| Temperature | 1.0 | 1.0 | 1.0 |
| Top-p | 0.9 | 0.9 | 0.9 |

Table 12: Settings for different LM agent backbones.

## A.8 VISUALWEBARENA AND OMNIACT SUBSET

We experimented with subsets of VisualWebArena and OmniACT to save on costs. We list them here for reproducibility.

For all VisualWebArena experiments we used the following:

```
[13, 15, 50, 129, 164, 167, 0, 77, 86, 89, 98, 99, 100, 101, 105,
130, 131, 142, 143, 146, 150, 189, 16, 29, 37, 38, 39, 47, 49, 52,
53, 56, 60, 61, 62, 69, 73, 76, 77, 81, 148, 173, 193, 196, 201,
212, 216, 231, 273, 314, 315, 322, 445]
```

For GPT-4v ordering ablations on OmniACT we used the following:

```
[4, 58, 115, 147, 156, 162, 165, 178, 179, 194, 204, 218, 235,
240, 248, 297, 353, 374, 391, 392, 395, 404, 409, 419, 434, 462,
487, 492, 517, 533, 556, 573, 598, 658, 667, 673, 678, 719, 795,
827, 896, 910, 944, 961, 975, 1018, 1025, 1038, 1084, 1093, 1101,
1103, 1128, 1130, 1138, 1142, 1147, 1181, 1192, 1219, 1252, 1284,
1291, 1353, 1427, 1442, 1448, 1514, 1521, 1538, 1580, 1590, 1594,
1600, 1606, 1622, 1636, 1641, 1665, 1684, 1694, 1696, 1710, 1711,
1719, 1726, 1731, 1740, 1743, 1845, 1877, 1883, 1918, 1924, 1951,
1960, 1993, 1994, 1997, 2011]
```

A.9    PROMPT

Listing 1: Language Model Prompt

```
You are an autonomous intelligent agent tasked with navigating desktop
    and web applications. You will be given tasks that can be
    accomplished by various actions that will be mapped to pyautogui code
    .

Here's the information you'll have:
The user's objective: This is the task you're trying to complete.
The current desktop screenshot: This is a screenshot of the desktop or
    webpage, with each interactable element assigned a unique numerical
    id. Each bounding box and its respective id shares the same color.
The observation, which lists the IDs of all interactable elements on the
    current screenshot with their text content if any, in the format [id]
     [tagType] [text content]. tagType is the type of the element. text
    content is the text content of the element. For example, [1234] [
    BUTTON] ['Add to Cart'] means that there is a button with id 1234 and
     text content 'Add to Cart' on the current web page. [] [StaticText]
    [text] means that the element is of some text that is not
    interactable.

The actions you can perform fall into two categories:

Mouse Actions:
click [id]: This action clicks on an element with a specific id.
double_click [id]: This action double clicks on an element with a
    specific id.
right_click [id]: This action right clicks on an element with a specific
    id.
hover [id]: Hover over an element with id.

Keyboard Actions:
type [content]: Use this to type content. Be sure to use other commands
    to click before or press enter after if necessary.
press [key_comb]: Simulates the pressing of a key combination on the
    keyboard (e.g., enter).
hotkey [key1] [key2]: Simulates the pressing of a multiple key
    combinations on the keyboard. For example, hotkey [Ctrl] [Alt] [
    Delete] will press Ctrl+Alt+Delete.

To be successful, it is very important to follow the following rules:
1. You should only issue actions that are valid given the current
    observation. Everything is possible. You MUST issue actions.
2. You can issue a sequence of actions separated by newlines.
3. You should follow the examples from past messages to reason step by
    step and then issue the next actions.
4. You should start every answer with "Let's think step-by-step"
5. Generate the actions in the correct format. Start with a "In summary,
    the next actions I will perform are" phrase, followed by the actions
    inside '''. Each action should be split by a newline. There should be
     no text inside '' except for the actions. For example, "In summary,
    the actions I will perform are '''click [1234] type [sample text]
    press [enter]'''".

Here are a few examples:
Example 1:

{Example 1}

Example 2:

{Example 2}
```

```
Example 3:

{Example 3}

Those were the examples. Now make a prediction given the observation.

OBSERVATION:

{Observation}
```

