# OpenReview forum: "The Impact of Element Ordering on LM Agent Performance"
_ICLR.cc/2025/Conference — Submitted to ICLR 2025_

### Official Review · Reviewer_UaCt · 2024-10-25

**Soundness:** 3
**Presentation:** 2
**Contribution:** 2
**Rating:** 5
**Confidence:** 3

**Summary:**

This manuscript investigates the influence of ordering (in which elements are presented) to the performance of language model agents on navigating virtual environments, and proposes two simple ordering methods (raster and t-SNE) that are better than random ordering.

**Strengths:**

1. The investigated problem (ie, ordering’s effects) is somewhat novel.
2. The findings that ordering can influence the effects are intuitive and consistent with human perception.
3. The paper is presented clearly and easy to follow.

**Weaknesses:**

W1. Some conclusions are not well supported by experimental results.
- 1.1 In L299, the manuscript claims that “Random ordering results in a similar performance drop to removing all HTML text descriptions”. However, random ordering results in 37.04% (Gemini 1.5) and 44.44% (GPT4-V) in Table 2, while removing all texts brings 3.7% (Gemini 1.5) and 38.89% (GPT4-V) in Table 1. The performance gap between these two is still large.
- 1.2 In the Introduction, the manuscript claims that the t-SNE ordering improves performance. However, in Table 4, t-SNE is worse than raster in (a) VWA with Llama3, and (b) OmniACT with human annotations. Here, 4 out of 9 metrics, almost half, show raster is better than t-SNE.

W2. The analyses and explanations of ordering’s effects are not sufficient. The manuscript notices that t-SNE is worse than raster in OmniACT with human annotations and provides these results in L455. However, why raster is better than t-SNE with human annotations has not been further analyzed.

W3. The necessity of using t-SNE to reduce dimension from 2D to 1D is not well stated. It is well known that t-SNE is more suitable for high-dimensional features, but the dimension of the original space is small (ie., 2D). In such case, using t-SNE will roughly equal the process that (1) choosing the top-left one element A, (2) finding its nearest neighbor element B, (3) then finding the nearest neighbor of both element A and element B (with a weighting strategy like exponential moving average), and so forth.

W4. The experimental settings are not clear.
- 4.1 In L704, the manuscript states that only a subset of OmniACT is used in experiments. However, in Sec. 6.2, the manuscript directly compares the results with previously reported results tested on the full OmniACT. This raises concerns about the comparison results.
- 4.2 In Table 5, the manuscript claims state-of-the-art performance. However, the sequence score is much lower than previous methods. Also, the foundational models are different. It is hard to say whether the performance difference is from the foundational models or the designs.

W5. There are some writing mistakes.
- In L212, the words “must find” are repeated two times.
- In L461, “on Omniact” should be “on OmniACT”.
- In L465, “a LM” should be “an LM”.

**Questions:**

I am not very familiar with this field, so I will adjust my rating after reading other reviewers' feedbacks.

Please see weaknesses W1 to W4. Writing mistakes do not need a response.

---

> ### Author Response · Authors · 2024-11-23
> **Reviewer UaCt Response**
>
> **Snippet: Some conclusions are not well supported by experimental results. 1.1 In L299, the manuscript claims that 'Random ordering results in a similar performance drop to removing all HTML text descriptions'. However, random ordering results in 37.04% (Gemini 1.5) and 44.44% (GPT4-V) in Table 2, while removing all texts brings 3.7% (Gemini 1.5) and 38.89% (GPT4-V) in Table 1\. The performance gap between these two is still large. 1.2 In the Introduction, the manuscript claims that the t-SNE ordering improves performance. However, in Table 4, t-SNE is worse than raster in (a) VWA with Llama3, and (b) OmniACT with human annotations. Here, 4 out of 9 metrics, almost half, show raster is better than t-SNE.**
>
> - In Table 1, removing all text means that no text at all is provided. In Table 2, we remove all “text descriptions” which is all of the *visible* text, but we retain tags such as \[IMG\].
> - We consider raster a potential heuristic to consider alongside t-SNE. If it becomes feasible to generate human GUI annotations at scale, (the scenario which OmniACT with human annotations considers), one should consider raster ordering. However, we consider predicted elements using our UI detection model as the more realistic current scenario.
>
> **Snippet: The analyses and explanations of ordering's effects are not sufficient. The manuscript notices that t-SNE is worse than raster in OmniACT with human annotations and provides these results in L455. However, why raster is better than t-SNE with human annotations has not been further analyzed.**
>
> - Our initial thought was that the difference was due to element count as human annotations have much less elements; however, our later analysis indicated that the effect of element counts is roughly the same between raster and t-SNE. We have included this additional analysis in section Appendix A.4. In this work, our goal was to show that ordering *drastically affects LM Agent performance*. Further investigation as to the mechanisms behind such agent behavior is an interesting direction for future work.
>
>
> **Snippet: The necessity of using t-SNE to reduce dimension from 2D to 1D is not well stated. It is well known that t-SNE is more suitable for high-dimensional features, but the dimension of the original space is small (ie., 2D). In such case, using t-SNE will roughly equal the process that (1) choosing the top-left one element A, (2) finding its nearest neighbor element B, (3) then finding the nearest neighbor of both element A and element B (with a weighting strategy like exponential moving average), and so forth.**
>
> - We do ***not*** claim that t-SNE is the optimal solution for element ordering. Instead, we aim to show that (1) ordering can be *impactful*, and (2) dimensionality reduction methods like t-SNE can *improve* orderings relative to reasonable baselines. The approach the reviewer described indeed sounds like a potential alternative to consider\!
>
> **Snippet: The experimental settings are not clear. 4.1 In L704, the manuscript states that only a subset of OmniACT is used in experiments. However, in Sec. 6.2, the manuscript directly compares the results with previously reported results tested on the full OmniACT. This raises concerns about the comparison results.**
>
> - In 4.1, we use a subset of Visual Web Arena, not OmniACT. In all of our final OmniACT experiments in Table 5, we use the full dataset.
>
> **Snippet: In Table 5, the manuscript claims state-of-the-art performance. However, the sequence score is much lower than previous methods.**
>
> - Sequence score is not an appropriate metric as it rewards actions on incorrect elements. It is possible to achieve a perfect sequence score and have a 0% task success rate; this is not true for action score. We describe the differences in detail in Table 5\. Accordingly, while we do include sequence scores for symmetry with OmniACT, we argue that action score is much more indicative of an agent’s overall performance.
>
> **Snippet: The foundational models are different. It is hard to say whether the performance difference is from the foundational models or the designs.**
>
> - We disagree and believe that our extensive ablations provide evidence that the performance improvements are due to our approach and not the difference in foundational models.  In Table 5 we provide a series of experiments that progressively show how our approach improves performance.

---

### Official Review · Reviewer_8p1Y · 2024-11-03

**Soundness:** 3
**Presentation:** 2
**Contribution:** 2
**Rating:** 5
**Confidence:** 4

**Summary:**

The paper explores the effect of element ordering on LLM agent performance in virtual environments like web and desktop. Through ablation studies on the VisualWebArena and OmniACT benchmarks, they find that random element ordering significantly degrades performance, similar to removing all textual information. To address this, the authors propose using dimensionality reduction techniques, specifically t-SNE, to create effective element orderings based on spatial relationships in pixel-based environments. The authors contribute a trained UI detection model and achieve state-of-the-art performance on OmniACT, demonstrating the impact of ordering strategies for agent navigation in pixel-only environments.

**Strengths:**

- Originality: The paper tackles a relatively unexplored area—optimizing UI element ordering for LM agents in pixel-only environments. The use of t-SNE for ordering based on spatial relationships is a novel application, offering a fresh perspective on improving agent navigation performance.
- Quality: The research employs ablation studies on VisualWebArena and OmniACT. The methodological depth and comparison across multiple ordering methods highlight the improvement of the approach.
- Clarity: The paper presents the problem and solution clearly, with intuitive descriptions of complex concepts like dimensionality reduction for ordering.
- Significance: Achieving state-of-the-art performance on OmniACT, the paper has practical implications for LM agent design in unstructured environments. The release of the trained UI detection model will further enhances its contribution to the research community.

**Weaknesses:**

- The paper could clarify its discussion of the dimensionality reduction approach, specifically addressing the parameters used in t-SNE. As t-SNE can be sensitive to parameter tuning, a detailed analysis of how parameter choices affect ordering outcomes would strengthen the validity of the results.
- Another weakness is the limited exploration of alternative ordering methods. While t-SNE provides performance gains, the study would benefit from a broader examination of ordering techniques, especially methods more aligned with semantic or functional groupings, which could be valuable in complex UI layouts where spatial proximity does not equate to functional relevance.
- While the authors release the trained UI detection model, the study would benefit from greater transparency in terms of training data diversity and model performance metrics, such as precision and recall for UI element detection, to better assess its effectiveness across domains.
- The ablation study lacks rigorous control over experimental variables, which makes it difficult to isolate the exact impact of individual attributes on agent performance. For instance, the experiments appear to vary multiple aspects of the state representation without consistently controlling. A more systematic approach to ablation studies would enhance the credibility of the findings.

**Questions:**

- The paper mentions using default parameters for t-SNE, but this method can be sensitive to parameter choices, such as perplexity and learning rate. Please provide a sensitivity analysis of key t-SNE parameters (e.g., perplexity, learning rate) and their impact on ordering performance.
- While t-SNE shows promising results for ordering, it may also be beneficial to explore other dimensionality reduction techniques like UMAP and MDS. Please compare t-SNE with at least one other dimensionality reduction technique (e.g., UMAP) on a subset of data, and report the relative performance in terms of agent task success rate.
- While the appendix provides training details for the UI detection model, the paper would benefit from additional performance metrics specific to this scenario. Please provide a breakdown of the training data composition and to report related object detection metrics (e.g., mAP, precision-recall curves) for the UI detection model.
- Tables 4 and 5 report key results on different ordering methods, yet they lack detailed context, such as specific configurations or experimental conditions for each row. Could the authors expand on these tables with additional descriptions or footnotes? For example, is the trained model used for UI detection consistent across all experiments, does the success rate and action score reflect the same metric on OmniACT, etc.

---

> ### Author Response · Authors · 2024-11-23
> **Reviewer 8p1Y Response**
>
> **Snippet: The paper could clarify its discussion of the dimensionality reduction approach, specifically addressing the parameters used in t-SNE. As t-SNE can be sensitive to parameter tuning, a detailed analysis of how parameter choices affect ordering outcomes would strengthen the validity of the results.**
>
> - In our experiments, we noticed that t-SNE is relatively robust to changes to the perplexity parameter. We have added these experimental results to section Appendix A.5.
>
>
> **Snippet: Another weakness is the limited exploration of alternative ordering methods. While t-SNE provides performance gains, the study would benefit from a broader examination of ordering techniques, especially methods more aligned with semantic or functional groupings, which could be valuable in complex UI layouts where spatial proximity does not equate to functional relevance.**
>
> - We do not claim that t-SNE is the optimal ordering technique and address this in more detail in our overall comments. Additionally, due the expensive costs associated with agent experimentation, we purposefully limited ourselves to t-SNE as a representative example of how dimensionality reduction can lead to improved orderings over reasonable baselines.
>
> **Snippet: While the authors release the trained UI detection model, the study would benefit from greater transparency in terms of training data diversity and model performance metrics, such as precision and recall for UI element detection, to better assess its effectiveness across domains.**
>
> - We thank the reviewer for their interest in our UI element detection model\! We have added significantly more detail in our revision (sections Appendix A.2 and A.3) and address this in more detail in our overall comments. We would be eager to add any additional details requested in our final camera-ready.
>
>
> **Snippet: The ablation study lacks rigorous control over experimental variables, which makes it difficult to isolate the exact impact of individual attributes on agent performance. For instance, the experiments appear to vary multiple aspects of the state representation without consistently controlling. A more systematic approach to ablation studies would enhance the credibility of the findings.**
>
> Our goal was to rigorously evaluate various aspects of the environment’s representation (i.e. element attributes) and unless specifically stated we kept the state representation consistent. In Tables 1 and 2 we changed the state representation specifically to ablate various element attributes. In Table 6, we changed only individual aspects of the state representation in all of our experiments (again to ablate the effects of individual changes). OmniACT uses a different state representation, but it is non-reproducible given that none of their code, trajectory, and state files are available.
>
> **We address Questions 1-3 above.**
>
> **Question 4: Tables 4 and 5 report key results on different ordering methods, yet they lack detailed context, such as specific configurations or experimental conditions for each row. Could the authors expand on these tables with additional descriptions or footnotes? For example, is the trained model used for UI detection consistent across all experiments, does the success rate and action score reflect the same metric on OmniACT, etc.**
>
> - Success rate and action score are equivalent in Table 4 to allow for comparison with Visual WebArena. In Table 5 we use OmniACT’s definition of action score.
> - In Table 5, the same trained UI detection is used for all rows except for Row 1 (which is taken from the OmniACT paper). In Table 4, the ordering methods are listed.

---

### Official Review · Reviewer_Ez7z · 2024-11-04

**Soundness:** 2
**Presentation:** 3
**Contribution:** 2
**Rating:** 5
**Confidence:** 5

**Summary:**

This paper investigates the impact of element ordering on the performance of language model agents. The empirical study highlights that the sequence in which UI elements are presented to the agent significantly influences performance, especially for elements represented solely by pixels, where traditional hierarchical ordering is unavailable.

The authors propose an ordering method based on dimensionality reduction, necessitating the training of a UI element detection model to extract elements from pixel data. This approach offers a viable solution for ordering in pixel-only environments. The proposed method consistently demonstrates performance improvements over random ordering and achieves a new state-of-the-art result on the OmniACT benchmark.

**Strengths:**

1. Reveals the Critical Impact of Element Order on Agent Performance: Through systematic experiments, the paper effectively demonstrates the significant influence of element ordering on language model agents operating in pixel-only environments. This finding presents a new perspective for developing efficient virtual environment navigation algorithms. Previous research has often concentrated on accuracy in image recognition and text analysis, overlooking the importance of element order in contextual understanding. This paper addresses this gap by introducing order optimization as a novel method for enhancing agent performance.

2. Provides an Ordering Method in Environments Lacking Hierarchical Information: In situations where structural information (e.g., HTML or DOM trees) is not available, traditional methods struggle to identify an optimal element order. By utilizing t-SNE for dimensionality reduction, the paper effectively maps pixel-level interface elements to a one-dimensional sequence, establishing a practical ordering method for pixel-only environments. Despite its simplicity, this approach achieves performance gains in pixel-based contexts, setting new performance benchmarks on platforms like OmniACT.

**Weaknesses:**

1. While the baseline of random ordering is understandable as detrimental to large language models in interpreting UI, it is overly simplistic and is not a strong baseline. The study would benefit from incorporating more heuristic baselines. For instance, could a vision-language model, such as GPT-4V, assist in determining an optimal ordering when seeing the UI directly?

2. In Table 6, the proposed method consistently outperforms other ordering techniques only when elements are detected using Faster R-CNN, failing to demonstrate consistent advantages in other scenarios.

3. Although the paper validates the effectiveness of various ordering methods (e.g., t-SNE ordering surpassing random ordering), it lacks a systematic analysis of why specific ordering methods enhance performance. The authors primarily illustrate the effects of ordering through experiments but do not investigate the mechanisms by which different strategies influence the language model's understanding and contextual construction.

**Questions:**

Please refer to the weaknesses section above for specific points that require clarification or further detail.

---

> ### Author Response · Authors · 2024-11-23
> **Reviewer Ez7z Response**
>
> **Snippet: While the baseline of random ordering is understandable as detrimental to large language models in interpreting UI, it is overly simplistic and is not a strong baseline. The study would benefit from incorporating more heuristic baselines. For instance, could a vision-language model, such as GPT-4V, assist in determining an optimal ordering when seeing the UI directly?**
>
> - We disagree that random ordering is overly simplistic. In fact, random ordering is often the default in existing GUI agent works. For example, in the well-cited GPT-4V in Wonderland (Yan, 2023) the ordering is clearly random in Figures 3 and 4 on pages 5 and 6 respectively. Our paper is the first GUI agent work that considers element ordering and we believe this benefits all future work in this field. We limited our work to three potential orderings as agent experimentation is extremely expensive.
> - Additionally, from our experience existing VLMs (vision-language models) are unable to parse GUIs at a fidelity level required for navigation. There have been very recent works that attempt to tackle this problem by training VLMs dedicated to GUI understanding, but that is beyond the scope of our work.
>
> **Snippet: In Table 6, the proposed method consistently outperforms other ordering techniques only when elements are detected using Faster R-CNN, failing to demonstrate consistent advantages in other scenarios.**
>
> - The scenario where elements are detected using Faster R-CNN is the most relevant and realistic scenario given that GUI human annotations are not scalable. We do not claim that t-SNE is the optimal ordering, simply that is the best ordering for a realistic workflow. We discuss more on this in our overall response.
>
> **Snippet: Although the paper validates the effectiveness of various ordering methods (e.g., t-SNE ordering surpassing random ordering), it lacks a systematic analysis of why specific ordering methods enhance performance. The authors primarily illustrate the effects of ordering through experiments but do not investigate the mechanisms by which different strategies influence the language model's understanding and contextual construction.**
>
> - We are mostly working with closed source models, so our ability to hypothesize on the mechanisms is limited. Our intuition is that the pre-training data for most LLMs contain HTML/Accessibility Trees that have elements ordered by pre-order traversal hence why pre-order ordering performs the best. Both raster and t-SNE lack the hierarchical nature that pre-order traversal allows for which is why there is such a gap. Bridging this gap is complex and requires further scrutiny by the research community.

---

### Official Review · Reviewer_4H1W · 2024-11-04

**Soundness:** 4
**Presentation:** 3
**Contribution:** 3
**Rating:** 6
**Confidence:** 5

**Summary:**

This paper investigates language model agents' navigation capabilities in virtual environments and focuses on exploring how element ordering affects agent performance. The paper's main finding is that element ordering has a significant impact on agent performance, with random ordering leading to substantial performance degradation - comparable to the effect of completely removing all visible text from web pages. While web-based agents can benefit from the semantic hierarchy provided by browsers, agents that parse elements directly from pixels cannot access this ordering information. The paper proposes an ordering method based on dimensionality reduction (such as t-SNE) that performs well in scenarios where only pixel information is available.

**Strengths:**

1. The paper provides the first in-depth investigation of how element ordering affects agent performance and demonstrates its significance.

2. The proposed dimensionality reduction-based ordering method performs well in pixel-only scenarios and achieves new state-of-the-art performance on the OmniACT agent benchmark.

3. The paper introduces a UI element detection model which has been made publicly available for other researchers to use and improve upon.

**Weaknesses:**

1. While the paper focuses on the t-SNE dimensionality reduction ordering method, it lacks in-depth analysis and comparison with other ordering methods. Additionally, all ordering methods show significant performance gaps compared to Pre-ordering (Table 4).

2. The paper briefly introduces the training process of the UI element detection model but lacks more detailed specifics.

**Questions:**

1. How does element ordering affect results at different orders of magnitude (e.g., fewer than 10 elements vs. more than 50 elements)?

2. While the ablation section examines how different inputs affect experimental results, to draw more convincing conclusions, the authors should conduct additional ablation experiments on datasets beyond VisualWebArena.

---

> ### Author Response · Authors · 2024-11-23
> **Reviewer 4H1W Response**
>
> **Snippet: While the paper focuses on the t-SNE dimensionality reduction ordering method, it lacks in-depth analysis and comparison with other ordering methods. Additionally, all ordering methods show significant performance gaps compared to Pre-ordering (Table 4).**
>
> - We analyze and compare t-SNE against the other methods across two benchmarks (Visual Web Arena and OmniACT). We have added more detail to the section Appendix A.4 (to answer your question on the effects of order of magnitude) in the revised version of the paper.
> - We do not claim that t-SNE is the optimal solution for element ordering and address this in our overall comments. The gap between the presented ordering methods and pre-ordering indicate potential for improvements and hopefully future research into this area.
>
> **Snippet: The paper briefly introduces the training process of the UI element detection model but lacks more detailed specifics.**
>
> - We thank the reviewer for their interest in our UI element detection model\! We have added more details in our revision (sections Appendix A.2 and A.3) and address this in more detail in our overall comments.
>
> **Question: How does element ordering affect results at different orders of magnitude (e.g., fewer than 10 elements vs. more than 50 elements)?**
>
> - If there are more elements, the gap between random vs either raster ordering and t-SNE increases (i.e. ordering is more important with more elements). However, the gap between raster and tSNE stays roughly the same. We have added this information to section Appendix A.4 in our revision.
>
> **Question: While the ablation section examines how different inputs affect experimental results, to draw more convincing conclusions, the authors should conduct additional ablation experiments on datasets beyond VisualWebArena.**
>
> - We test our hypothesis on multiple benchmarks: Visual Web Arena which focuses on web environments and OmniACT which focuses on desktop environments. OmniACT itself is an incredibly diverse dataset, containing over 2000 tasks and 38 different desktop applications across Mac OS, Windows, and Linux. We show through extensive experiments that ordering plays an important role in both environments. We believe that our experimental results are sufficient to indicate the importance of ordering, but are eager to explore other benchmarks in future work.

---

> > ### Comment · Reviewer_4H1W · 2024-12-02
> > **Official Comment by Reviewer 4H1W**
> >
> > Thank you for your response. I will maintain my rating.

---

### Meta-Review · Area_Chair_nAF4 · 2024-12-23

**Metareview:**

This paper reveals that the ordering of UI elements significantly impacts language model agent performance in virtual environments - randomly ordering elements reduces performance as much as removing all visible text. The authors propose using dimensionality reduction (specifically t-SNE) to create effective element orderings in pixel-based environments, achieving state-of-the-art results on the OmniACT benchmark with more than double the task completion rate compared to previous approaches.

While the paper introduces an important consideration about element ordering in LM agent performance, it has several key limitations. The core claims about random ordering effects are not well-supported by data, and the proposed t-SNE approach shows mixed results compared to simpler baselines. The paper lacks thorough analysis of why different orderings work or fail, and the evaluation methodology makes it difficult to isolate the impact of the ordering approach. Despite raising interesting questions, the work needs substantial revision to strengthen its scientific contributions. Besides, the authors accidentally revealed their identity during the rebuttal (but the decision would be the same regardless).

**Additional Comments On Reviewer Discussion:**

See above.

---

### Decision · Program_Chairs · 2025-01-22

Reject